# Immunogenicity of an Intranasal Dual (Core and Surface)-Antigen Vaccine Against Hepatitis B Virus Enhanced by Carboxyl-Vinyl Polymer Excipients

**DOI:** 10.3390/vaccines13050464

**Published:** 2025-04-25

**Authors:** Md Haroon Or Rashid, Fumihiko Yasui, Takahiro Sanada, Risa Kono, Tomoko Honda, Bouchra Kitab, Lipi Akter, Masashi Utsunomiya, Risa Sato, Osamu Yoshida, Yoichi Hiasa, Yasunori Oda, Yasumasa Goh, Takashi Miyazaki, Michinori Kohara, Kyoko Tsukiyama-Kohara

**Affiliations:** 1Joint Faculty of Veterinary Medicine, Kagoshima University, Kagoshima 890-0065, Japan; haroon9330@gmail.com (M.H.O.R.); bouchra.kitab17@gmail.com (B.K.); lipi.cvasu.44@gmail.com (L.A.); masashi0622u@outlook.jp (M.U.); urichan3636@gmail.com (R.S.); 2Department of Microbiology and Cell Biology, Tokyo Metropolitan Institute of Medical Science, Tokyo 156-8506, Japan; yasui-fm@igakuken.or.jp (F.Y.); sanada-tk@igakuken.or.jp (T.S.); kono-rs@igakuken.or.jp (R.K.); honda-tm@igakuken.or.jp (T.H.); kohara-mc@igakuken.or.jp (M.K.); 3Department of Gastroenterology and Metabology, Ehime University Graduate School of Medicine, Toon 791-0295, Japan; Yoshida.osamu.ny@ehime-u.ac.jp (O.Y.); hiasa@m.ehime-u.ac.jp (Y.H.); 4Beacle Co., Ltd., Kyoto 606-8305, Japan; oda@beacle.com (Y.O.); y_goh@beacle.com (Y.G.); 5Toko Yakuhin Kogyo Co., Ltd., Toyama 930-0211, Japan; t.miyazaki@toko-yakuhin.co.jp

**Keywords:** HBV, CVP, hy-LHBs, HBc

## Abstract

**Background:** Hepatitis B virus (HBV) is a major cause of morbidity and mortality globally, and chronic infections are associated with cirrhosis and hepatocellular carcinoma. Issues with conventional treatments and vaccines mean there is a need for new therapeutic vaccines, which must elicit a strong and sustainable immune response. Here, we evaluated the immunogenicity of dual-antigen vaccines containing hybrid surface (hy-LHBs) and core (HBc) antigens, combined with a carboxyl-vinyl polymer (CVP) as a mucoadhesive excipient, following intranasal administration in mice. **Methods:** Mice were intranasally administered a mixed vaccine (10 µg of hy-LHBs and 2.5 or 10 µg of HBc) with or without a CVP excipient, and they were assessed for their immune response (levels of IgGs or IgA antibodies in an ELISA, IFN-γ level in splenocytes in an ELISpot assay, and cytokine/chemokine levels in a BioPlex assay). A protein stability assay was also conducted for vaccine formulations with and without excipients. **Results:** Significantly enhanced IgG production was noted targeting hy-LHBs and (less markedly) HBc at 10 µg/antigen, but only a non-significant elevation was noted with the vaccine containing 2.5 µg HBc. The BioPlex assay showed a significant increase in IL-2 (#00-07, 0B), IL-12(p40)(#00), eotaxin (#00), MIP1α (#00, #00-07, 0B), and MCP-1 (#00-07, 0B) in mice that received treatment compared to those of untreated mice. The endpoint titers of IgG1 and IgG2a were measured, which were higher with CVP excipients than without. From the IgG2a/IgG1 ratio, a higher IgG1 response was induced by CVPs to hy-LHBs and a higher IgG2a response was induced to HBc. Th2-dominant phenotype to hy-LHBs was induced with CVP#00 in an ELISpot assay. The highest anti-hy-LHBs antibody titer was noted with the conventional CVP#00 excipient. Consistent with these results, a higher amount of neutralizing antibodies of HBV was induced with CVP#00 treatment and followed by #00-03 and #14-00. **Conclusions:** We consider that the addition of CVP excipients to vaccine formulation enhances immunogenicity and HBV antigen stability for intranasal vaccines. This effect was seen for both humoral and cell-mediated immune responses, indicating the potential of CVPs as excipients in intranasal HBV vaccines.

## 1. Introduction

Hepatitis B virus (HBV) infection remains a major global public health challenge, despite the availability of approved therapeutics and vaccines for over three decades. Recent estimates indicate that approximately 260 million people worldwide are chronically infected with HBV, with the vast majority of cases remaining undiagnosed and untreated [1,2]. Chronic HBV infection increases the risk of severe liver diseases, including cirrhosis, hepatic decompensation, and hepatocellular carcinoma, with 15% to 40% of affected individuals developing these complications over their lifetime [3,4].

Chronic HBV infection can be treated with nucleotide analogs (the most commonly prescribed therapy), which can effectively inhibit viral reverse transcriptase and suppress HBV replication to undetectable levels [5,6]. However, there are issues with the development of virus resistance to therapy and adherence to a treatment regimen which must, of necessity, be lifelong. Thus, much emphasis in research on HBV has focused on vaccines. One potential issue with this approach is that chronic HBV infection impairs the host immune response, particularly the T-cell response, which may reduce efficacy for conventional vaccines [7,8]. The search for more potent vaccines to overcome such immunological hypo-responsiveness or non-responsiveness continues [9,10,11].

One proposed solution involves the intranasal administration of dual-antigen therapeutic vaccines. Nasal vaccination is regarded as a useful administration route for activating mucosal T cells, as both mucosal and systemic immune responses may occur when antigens are present in the nasopharynx-associated lymphoid tissue. Furthermore, vaccines containing surface antigens (HBs antigens) as well as core antigens (HBc antigens) have the advantage of presenting more epitopes; such vaccines may induce broad-spectrum immunity in both the mucosal and systemic compartments [12]. One vaccine against HBV developed with this approach is the HeberNasvac^®^ intranasal formulation, which elicited antibodies clinically in preliminary studies, and it has demonstrated superior efficacy to PEGylated interferon treatment in chronic hepatitis B (CHB) treatment-naïve patients [13]. Although promising results have thus been demonstrated for therapeutic vaccines with this approach, the challenge of completely eradicating the virus remains. It is thus important to develop more potent therapeutic vaccines that will allow patients to completely eliminate the virus from their bodies. Building on the success demonstrated by HeberNasvac^®^, antigen configuration and vaccine delivery would appear to offer two paths to enhanced potency [13].

In terms of HBV surface proteins, the HBs-large (LHBs) antigen has received attention. It consists of three domains (pre-S1, pre-S2, and S domains). A hybrid HBs-L antigen (hy-LHBs) has been developed by combining HBs-L proteins from both the HBV C and D genotypes, and a therapeutic vaccine based on this hybrid antigen elicited antibodies in mice, indicating its potential as a candidate vaccine [14]. Furthermore, the pre-S1 domain of the LHBs antigen has been implicated in the HBV infection of tree shrew and human hepatocytes through interactions with the host cellular receptor, the sodium taurocholate co-transporting polypeptide (NTCP), providing more evidence that this antigen is a potentially promising target for therapeutic antibodies [15].

In terms of vaccine delivery, the use of carboxyl-vinyl polymers (CVPs) as excipients has received much attention from researchers. A CVP can be added to spray gels to act as a mucoadhesive excipient, and it enhances the stability and durability of the intranasal vaccine and reduces leakage from the tissues, to which an intranasal vaccine is administered [14,16,17,18]. A CVP formulation has been approved for use as a vaccine excipient in Japan and has shown promise as a mucoadhesive agent for intranasal influenza vaccines. CVP increased nasal retention time and improved mucosal antibody responses in animal models without promoting antigen redirection to the central nervous system. Immunogenicity has been characterized for HBc and LHBs antigens when delivered intranasally with a CVP excipient [14,17]. Interestingly, findings in a clinical trial in phase II indicate that the HeberNasvac^®^ therapeutic vaccine may more effectively reduce viral DNA levels when it is combined with a CVP excipient for intranasal immunization [19].

Accordingly, in this study, we aimed to investigate the immunogenicity of dual-antigen experimental vaccines (HBc and hy-LHBs antigens) with CVP excipients by evaluating immune responses (both humoral and cellular responses) in BALB/c mice receiving a combination of HBc and LHBs antigens in the presence or absence of a CVP excipient, intranasally. We also investigated protein stability in an experimental vaccine formulation.

## 2. Materials and Methods

### 2.1. Ethics Statement

This study was carried out by the Japanese Association for Laboratory Animal Science’s Guidelines for Animal Experimentation and the National Institutes of Health’s Guide for the Care and Use of Laboratory Animals, it and was approved by the research ethics committee of Kagoshima University (Approval No. VM23025).

### 2.2. Animals

Purpose-bred (SPF), six-week-old BALB/c mice were purchased from Japan SLC, Inc. (Shizuoka, Japan), and housed in an experimental animal facility at Kagoshima University (Kagoshima, Japan). A total of thirty-two healthy female mice were used, with four mice per group in this study. The animal experiment was approved by the Animal Experimental Committees of Kagoshima University on 7 June 2021 (Permission No. 21-009). The experimental animal center at Kagoshima University is accredited by AALAC International (Accreditation No. 001698).

### 2.3. Cell Cultures and Viruses

HepG2-hNTCP-30 cells were obtained and cultured as described previously [17]. The original HepG2 cell lines were purchased from the America Type Culture Collection.

Virus stock of HBV genotype C (C_JPNAT; GenBank Accession No.: AB246345.1) was propagated in primary human hepatocytes derived from chimeric mice with human liver (PXB-cells; PhoenixBio, Higashi-Hiroshima City, Japan) as described previously [20].

### 2.4. Experimental Vaccines

CVP formulations were obtained from Toko Yakuhin Kogyo Co., Ltd., Osaka, Japan. The CVP and protein solutions were mixed to prepare the experimental vaccine formulations just before administration. The experimental vaccines used in this study are listed below. hy-LHBs and HBc proteins were purchased from Beacle, Inc., Tokyo, Japan.
**Experimental vaccine formulations**hy-LHBs 10 µg + HBc 10 µg, combined with #00 CVP formulation (carboxyl group [COOH] > 60.5% of base content, high molecular weight [>1,000,000]) viscosity, 100~120 mPa·s, finalhy-LHBs 10 µg + HBc 10 µg, combined with #00-03 CVP formulation with 0.02% cetylpyridinium hydrochloride (CPC) addedhy-LHBs 10 µg + HBc 10 µg, combined with #14-00 CVP formulation (highly polymerized poly-acryl acid, COOH base content > 61.7%)hy-LHBs 10 µg + HBc 10 µg, combined with #00-07 CVP formulation (0.4% poly solvate 80, 0.2% CPC)hy-LHBs 10 µg + HBc 10 µg, combined with #0B CVP formulation (high viscosity, 500–600 mPa·s, final), Hy-LHBs 10 µg + HBc 10 µg,hy-LHBs 10 µg + HBc 10 µg without CVP formulationhy-LHBs 10 µg + HBc 2.5 µg combined with #00 CVP formulationhy-LHBs 10 µg + HBc 2.5 µg without CVP formulation

### 2.5. Immunization and Immunization Schedule

The mice were administered with an experimental vaccine, with or without an equal volume of excipient, intranasally (dose of 7 µL per nostril) using an SP10 polyethylene tube and an Itoh micro-syringe (ITO CORPORATION., Shizuoka, Japan). The intranasal immunizations were performed bilaterally, with equal volumes administered to each nostril (7 µL dose × 2).

The day mice underwent the initial intranasal immunization was designated as day 0 (in week 0), and they received a second intranasal immunization at week 4.

Blood samples were collected from the orbital vein of each mouse at weeks 0 (immediately prior to immunization), 1, 2, 3, and 4 (immediately prior to immunization). At week 5, mice were euthanized under anesthesia by collection of whole blood from the heart via cardiac puncture. Body weight and temperature were also recorded for each mouse at the time of each blood collection. The study schedule is summarized in Figure 1.

(1)hy-LHBs antigen (Ag) 10 µg and HBc Ag 10 µg plus CVP①~⑥;(2)hy-LHBs Ag 10 µg and HBc Ag 2.5 µg plus CVP #00(⑦) and without CVP(⑧) CVP-

Antigen solution was mixed with a CVP excipient to prepare a vaccine formulation for intranasal immunization.

### 2.6. Detection of Antibodies Using Enzyme-Linked Immunosorbent Assay (ELISA)

To evaluate elicited HBc and hy-LHBs-antigen-specific antibody production, U96-Nunc-Maxisorp plates (Thermo Fisher Scientific, Waltham, MA, USA) were coated with 50 µL of hy-LHBs and HBc proteins, each diluted to 2 µg/mL in 50 mM carbonate buffer (pH 9.6). The phosphate-buffered saline (PBS) containing 1% bovine serum albumin, 0.5% Tween 20, and 2.5 mM ethylenediaminetetraacetic acid) was added to each well to block nonspecific binding. The plates were then incubated at room temperature for two hours. After blocking, individual mouse serum samples were diluted in blocking buffer and dispensed into the wells at 50 µL per well. The plates were incubated at room temperature for two hours to allow for the binding of antibodies. Following serum incubation, the plates were washed three times with PBS-containing 0.05% Tween-20 (PBST), followed by the addition of 50 µL of anti-mouse IgG or IgA rabbit IgG conjugated to horseradish peroxidase (HRP) (DAKO, Santa Clara, CA, USA) and diluted to 1 µg/mL in blocking buffer to each well. The plates were incubated at room temperature for one hour. After further incubation, the plates were washed six times with PBST, followed by the addition of 100 µL of TMB substrate solution containing hydrogen peroxide (SIGMA-ALDRICH, Burlington, MA, USA) to each well. The plates were then incubated at room temperature for 10 min to allow the color reaction to develop. The reaction was stopped by adding 50 µL of 2 M sulfuric acid to each well. Absorbance was measured at 450 nm (TMB substrate) using a microplate reader.

The production of IgG1 and IgG2a specific to HBc and hy-LHBs-antigen in the sera of mice was measured by ELISA using the above plates. The sera of mice were diluted serially by two-fold with a dilution buffer [1% BSA/0.5 mM EDTA/0.5% Tween 20/PBS(-)] and added to the wells. After incubation at 4 °C overnight, the wells were washed and incubated for 2 h at room temperature with 50 μL of HRP-conjugated goat anti-mouse IgG1 (1:10,000, Bethyl, A90-105P) or HRP-conjugated goat anti-mouse IgG2a (1:10,000, Bethyl, A90-107P) antibodies. After washing the wells, 100 μL of a mixture of TMB substance (BCL-TMB-21, Beacle) was added to each well. The reactions were quenched by adding 50 μL of 2 M sulfuric acid to each well, and the absorbance was measured at 450 nm. To calculate the ratio of IgG2a/IgG1 specific to HBc or hy-LHBs in the sera of vaccinated mice, endpoint titers of IgG2a and IgG1 specific to HBc or hy-LHBs were determined. The endpoint titer was defined as the reciprocal of the highest dilution of serum at which the absorbance at 450 nm exceeded two-fold the value of the blank.

### 2.7. Neutralization Test

Neutralization tests were performed as previously described [14]. Briefly, two-fold serial dilutions of the mouse sera in U-bottomed microplates were incubated for 1 h at 37 °C with an equal volume of HBV inoculum containing 3.75 × 10^5^ viral DNA copies, and 125 μL of each mixture was used to inoculate a HepG2-hNTCP-30 cell culture (5.0 × 10^4^ cells, 7.5 genome equivalents per cell). After incubation for 3 h on ice, the cells were washed 5 times with culture medium and collected. The resulting cell samples were used for quantitative PCR analysis as described previously. The neutralizing antibody titer of each serum sample was determined as the reciprocal of the maximum dilution of serum that reduced the viral DNA level by 90% or more compared to the control samples.

### 2.8. Multiple Cytokine Expression Analysis

The mouse sera were analyzed using the Bio-Plex Suspension Array System, which employs Luminex-based technology. A Mouse Cytokine/Chemokine Magnetic Bead Panel (23-plex: #M60009RDPD, 9-plex: #MD000000EL) was utilized following the manufacturer’s instructions (Bio-Rad, Hercules, CA, USA).

### 2.9. ELISpot Assay

Interferon-gamma (IFN-γ) and IL-4 production was quantified in splenocyte culture using an ELISpot assay as previously described [21,22] with slight modifications. Briefly, thawed splenocytes [3 × 10^5^ splenocytes/well] were seeded into the well of 96-well MultiScreen IP Sterile plates (Merck Millipore Ltd., Burlington, MA, USA) precoated with anti-IFN-γ (MABTECH, Cincinnati, OH, USA, Nacka strand, Sweden #3321-2H) or -IL-4 (MABTECH, Nacka strand, Sweden #3311-2H) antibody. HBV-specific T-cell responses were assessed by stimulating the splenocytes with either PepMix HBV (Large envelope protein) Ultra (JPT Peptide Technologies GmbH, Berlin, Germany) or PepMix HBV (Capsid Protein) Ultra (JPT Peptide Technologies GmbH) at 1 μg/mL. IFN-γ or IL-4 spot-forming cells (SFC) were counted using an ELISpot assay kit (MABTECH) according to the manufacturer’s instructions. After drying the ELISpot plates, the number of spots in each well was counted using an automated ELISpot plate reader (Advanced Imaging Devices GmbH, Strasberg, Germany). SFC values were indicated as the mean ± standard deviation.

### 2.10. Protein Stability Assay and Western Blot Assay

Vaccine formulation was assessed for the protein stability of the HBc and hy-LHBs antigens (Beacle Co., Kyoto, Japan). hy-LHBs antigen (10 μg/mL) was mixed with 0, 2.5, 5, or 10 μg/mL of HBc antigen and further mixed with an equal volume of CVP in a Protein LoBind tube (Eppendorf Co., Hamburg, Germany). These tubes were incubated at 4 °C for one week. Their protein stability was examined at 37 °C using a standard western blotting (WB) method with a 12% SDS-PAGE gel together with a Page Ruler pre-stained molecular weight marker (Thermo Fisher Scientific). The gel was electronically transferred to a nylon membrane (Immobilon-P, Millipore), blocked with 1% Block Ace in PBST, and reacted with anti-HBc rabbit polyclonal antibody (ab115992, Abcam Co., Cambridge, UK) or anti-HBs mouse monoclonal antibody (MoAb) (MA1-19263, Thermo Fisher Scientific). The chemiluminescence from the membrane was detected using the FUSION Solo chemiluminescence imaging system (Vilber, Marne-la-Vallee cedex 3, France).

### 2.11. Statistical Analysis

Data were analyzed statistically using one-way analysis of variance (ANOVA) and Dunnet’s Test with GraphPad Prism analytical software (Version 9). P values less than 0.05 were regarded as significant and are indicated.

## 3. Results

### 3.1. CVP Excipients Enhance Immunogenicity of Experimental HBc and hy-LHBs Antigen Vaccines

To investigate the immunogenicity of experimental vaccines containing HBc and HBs antigens and a CVP excipient, we evaluated antibody responses to a range of HBc plus HBs vaccine formulations with and without CVP excipients in intranasally immunized BALB/c mice (Figure 1).

At two weeks after the initial intranasal immunization at 10 μg/antigen (Figure 2A), experimental vaccines with CVP excipients elicited significantly greater antibody responses than the vaccines without excipients, for both the HBc antigen (#00-03 CVP excipient vs. vaccine without CVP; *p* value = 0.0258) and the hy-LHBs antigen (#00, #00-03, #14-00, #00-07, #0B CVP excipients vs. vaccine without CVP; *p* value = 0.001 to 0.0042).

A similar pattern was observed in analyses at weeks 3 and 4 after the initial immunization for hy-LHBs antigen with CVP vs. without CVP (Figure 2B,C). Responses to the HBc antigen were greater in the presence of CVP than its absence, but this was an absolute only and it was not statistically significant.

The greatest antibody response was elicited by the vaccine with the conventional CVP excipient (type CVP#00) followed by the vaccine with additional CPC (Type #003).

The experimental vaccine at 10 μg for the hy-LHBs antigen and 2.5 μg for the HBc antigen, with a CVP excipient, elicited a greater antibody response than the equivalent vaccine without CVP excipient at weeks 2, 3, and 4, but these differences were not significant. After 5 weeks post-first immunization, anti-HBc antibody production was also increased without CVP showing a significant difference with CVP, and anti-hy-LHBs antibody showed a significant increase with CVP#00 compared to that without CVP (Figure 2D). anti-HBc IgA and anti-hy-LHBs IgA showed higher reactions than those without CVP (Figure 2E).

To further evaluate antibody response, we subjected serially diluted sera obtained from the antigen-CVP-immunized mice at week 2 for ELISA evaluation (Figure 3). For anti-HBc IgG (Figure 3A), the highest reactivity and dilution ratio was observed for the conventional CVP excipient with added CPC (type CVP #00-03), followed by the conventional CVP excipient (type CVP#00), and that with poly solvate (type CVP#00-07). For anti-hy-LHBs IgG (Figure 3B), the highest significant reactivity and dilution ratio was observed for the conventional CVP excipient (type CVP#00, *p* = 0.0224), followed by the CVP excipient with poly solvate (type CVP#00-07, *p* = 0.047), and the CVP excipient with added CPC (type CVP#00-03, *p* = 0.1616) when compared to vaccines without a CVP excipient. Our results suggest that CVP excipients (especially types CVP#00 and CVP#00-03) enhance the immunogenicity of HBc and hy-LHBs experimental vaccines, considering the signs of stronger antibody responses in immunized mice.

The HBV-neutralizing Ab was measured, as previously reported [14].

Following intranasal immunization with hy-LHBs, HBc, and the CVP mix, anti-HBV neutralizing (NT) Ab was detected after 5 weeks (Figure 4). As a result, with CVP#00, #00-03, and #14-00, NT Ab was produced, but with CVP#00-07, #0B and CVP-, NT Ab was not detected.

### 3.2. Cytokine, Chemokine Levels and Th1/Th2 Responses

To investigate the effect of using CVP excipients on the wider immune response to experimental vaccines, we measured cytokine and chemokine levels using the BioPlex assay in immunized mouse sera at 2 weeks (Figure 5).

We observed a significant increase in IL-12(p40), eotaxin, and MIP-1α in treatment containing HBc and hy-LHBs antigens with CVP#00 compared to the untreated control (none, Figure 5). Similarly, CVP#00-07 induced significant increase in IL-2, IL-9, MCP-1, and MIP-1α, while CVP#0B led to significant increase in IL-2, MCP-1, and MIP-1α. Treatment with 10 μg of hy-LHBs and HBc, or 10 μg of hy-LHBs and 2.5 μg of HBc, showed a significant increase in MIP-1β.

We next investigate the Th1/Th2 balance against hy-LHBs and HBc antigen (Figure 6). The level of IgG1 was increased by CVPs, and significant increases against hy-LHBs with CVP#00 and HBc with CVP#00-03 were observed (Figure 6A). The level of IgG2a was increased with CVPs compared to without CVP (CVP-) (Figure 6B). The ratio of IgG2a/IgG1 indicated a Th1-dominant phenotype against hy-LHBs (#0B) and HBc (all) with CVP excipients (Figure 6C,D).

The stimulation of HBs or HBc peptides in splenocytes from mice immunized with hy-LHBs and HBc Ag with CVP#00, #14-00, and #0B was detected by ELISpot assay (Figure 6). Treatment with CVP resulted in higher IFNγ and IL-4 production than without CVP (CVP) (Figure 6E,F).

### 3.3. Effect of CVP on Stability of hy-LHBs and HBc Antigens

To assess the stability of the two antigens (hy-LHBs and HBc) in experimental vaccines with a CVP excipient, we mixed hy-LHBs antigen (10 μg/mL) with varying amounts (0–10 μg/mL) of HBc antigen, with or without added CVP (conventional formulation, type #00), and stored the mixtures at 4 °C for one week (Figure 7A, Appendix A). Each stored formulation was then sampled, and the samples were submitted to western blot analysis to ascertain the presence of the two antigens. Representative blots are shown in Figure 7. We obtained bands at the target sizes of approximately 42 kDa for the hy-LHBs antigen and 21 kDa for HBc. hy-LHBs (10 μg/mL) and HBc (0~10 μg/mL) were lower in the absence, than the presence, of CVP. Moreover, the hy-LHBs antigen level was highest with HBc 5 μg/mL followed by HBc 10 μg/mL and HBc 2.5 μg/mL. The stability of hy-LHBs and HBc antigens (10 μg/mL, respectively) was examined by incubation at 37 °C for 3, 6, 12, and 24 h with or without CVP#00 (Figure 7B, Appendix A). As a result, the mixture of CVP#00 increased the stability of both hy-LHBs and HBc antigens.

## 4. Discussion

In the present study, we examined the ability of dual-antigen (HBc and hy-LHBs) experimental vaccines with CVP excipients to elicit enhanced immune responses in intranasally immunized BALB/c mice. To the authors’ knowledge, this is the first report on responses elicited by intranasal vaccine formulations combining a hybrid HBV surface antigen (hy-LHBs), HBc, and CVP excipients.

For key findings in this study, we found that vaccines with CVP excipients elicited significantly enhanced antibody responses relative to those without excipients for both antigen-specific IgGs, suggesting that the addition of CVP may enhance vaccine immunogenicity and lead to a boosted immune response. IFN-γ levels were also increased in splenocytes from mice receiving a vaccine with the conventional CVP excipient (formulation #00) as assessed by the ELISpot assay consistent with a previous report [16]. Furthermore, the modification of cytokine and chemokine levels through the addition of CVP excipients may help this enhancement. A significant increase in IL-12(p40), Eotaxin, and MIP-1α, observed with CVP#00, was previously reported in splenic T cell activation by LPS treatment [23]. Similarly, CVP#00-07 also led to a significant increase in IL-2, IL-9, MCP-1, and MIP-1α in platelet-rich plasma [24] or in old individuals [25]. A significant increase in IL-2, MCP-1, and MIP-1α with CVP#0B was observed in the case of LPS treatment [23], aging [25], COVID-19 [26], NASH patients [27], and so on. However, only HBV antigen immunization (hy-LHBs + HBc) significantly increased MIP-1β, which was not observed in HBV antigen with CVP groups. Therefore, MIP-1β might contribute to the efficacy of CVPs. The combination of HB antigens enhanced the stability of the antigen mixture, leading to improved vaccine performance. Our findings are consistent with previous reports on CVP excipients increasing the viscosity and stability of vaccine formulations [17,28]. The significantly enhanced antibody production in excipient-receiving mice (for both antigens) may indicate that CVP conferred greater viscosity on vaccine formulations, which has been linked to more robust immune responses. In the current study, we found some evidence of an enhanced immune effect and the stability of antigens, based on serum antibody levels, suggesting immune effects extending beyond the mucosal compartment.

All CVP excipients appeared to be associated with higher antibody titers, but we found the highest titers for the conventional CVP formulation (type #00 CVP) and the CVP formulation with greater CPC content and molecular weight (type #00-03 CVP). Specifically, the former proved most effective in eliciting anti-hy-LHBs IgG, whereas the latter was more effective at eliciting anti-HBc IgG. This differential in efficacy highlights the potential for adapting CVP formulations based on the target antigen, which could be crucial for designing vaccines that aim to activate both arms of the immune system.

Our study also presents interesting evidence on the duration of vaccine effects, with antibody responses being sustained over a longer period (3–4 weeks) in excipient-receiving mice, suggesting that CVP not only enhances the initial immune response but also contributes to the longevity of the immune activation. This is consistent with the known stabilizing properties of CVPs, which prevent antigen degradation and ensure that antigens remain immunologically active for prolonged durations [29]. The results of our protein stability assays also suggest that the presence of CVP excipients helps to preserve the integrity of the HBc and hy-LHBs proteins within the experimental vaccine, ensuring that the complex remains intact and recognizable by the immune system over time.

We also report interesting findings in terms of the cellular immune response, in addition to those on the humoral immune response. The ratio of IgG2a/IgG1 of#0B was higher than other CVPs, suggesting a higher Th1-dominant phenotype to hy-LHBs, and that of #00 and #00-03 indicated a higher IgG1, suggesting a Th2-dominant phenotype against hy-LHBs or HBc. As stated above, splenocyte IFN-γ and IL-4 levels were higher in mice receiving the dual-antigen vaccine combined with CVP excipients. This finding is critical, as IFN-γ and IL-4 are known for their role in driving Th1 and Th2 responses, respectively, contributing to a balanced and effective immune response that includes both antibody-mediated and T-cell-mediated pathways [30,31]. Therefore, CVP#0B preferably induced the cellular immune response to hy-LHBs, and CVP#00 and #00-03 predominantly induced the humoral immune response to hy-LHBs. Consistent with this result, the immunization of hy-LHBs and HBc with CVP#00 and #00-03 strongly induced the NT antibody against HBV. By contrast, we did not find any difference in cytokine or chemokine levels, which suggests that the effect of CVP excipients on the immune system may not be an entirely direct effect. This view is consistent with a report that CVP excipient augmented the efficacy of an intranasal influenza vaccine in mice through facilitating nasal retention time rather than activating the immune system [17].

The differential effects of CVP formulations on immune activation also tie into this excipient’s mechanism of action. CVP has been shown to improve antigen delivery to antigen-presenting cells (APCs), a key step in initiating both T-cell and B-cell responses [32]. By ensuring efficient delivery and improved uptake of antigens, CVP enhances immune cell activation, which provides a plausible explanation for the stronger immune responses observed in mice receiving vaccines with CVP excipients. Furthermore, the multivalent nature of CVP formulations likely facilitates more effective B-cell receptor cross-linking, leading to enhanced activation and subsequent antibody production [33].

Our study also highlighted an important limitation regarding antigen stability when only hy-LHBs was present in the vaccine formulation. In the absence of HBc antigen, the stability of hy-LHBs was reduced, even in the presence of CVP. This finding suggests that the presence of both antigens at an appropriate ratio may be necessary for optimal immunogenicity preservation by the formation of the desired higher order structure, which has implications for vaccine formulation strategies. Combining both HBc and hy-LHBs antigens may enhance both immunogenicity and the stability of the antigen mixture, thus ensuring more effective vaccine performance.

In conclusion, CVP excipients appear to enhance the humoral and cellular immune responses to experimental vaccines containing HBc and hy-LHBs antigens in intranasally immunized mice. Furthermore, CVPs also appear to enhance protein stability for the two antigens in these vaccine formulations. The design of CVP formulations can be optimized to target specific immune responses, further improving the therapeutic potential of vaccines [34]. Future research should focus on elucidating the precise mechanisms through which CVP enhances antigen presentation and immune activation, as well as refining CVP formulations for optimal use in clinical applications.

## Figures and Tables

**Figure 1 vaccines-13-00464-f001:**
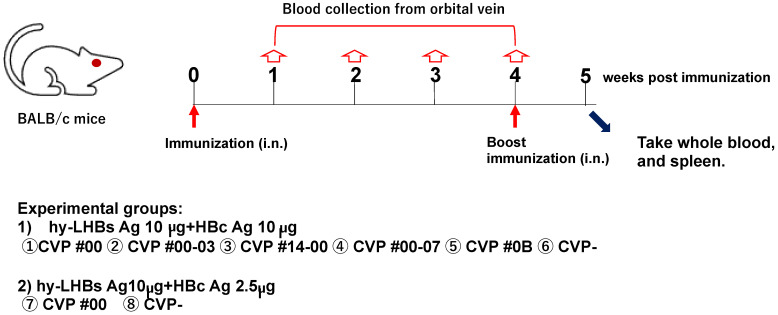
Schematic study schedule.

**Figure 2 vaccines-13-00464-f002:**
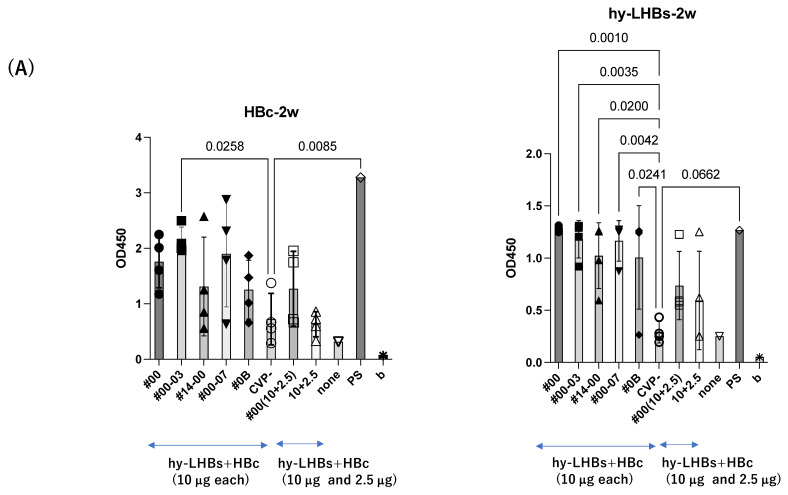
Detection of anti-HBc Ab (left) and anti-hy-LHBs (right) in mouse sera (×100) after 2 weeks (**A**), 3 weeks (**B**), 4 weeks (**C**), 5 weeks (**D**), and 5 weeks of IgA (**E**) of immunization by ELISA. CVP- shows hy-LHBs and HBc Ag 10 μg each without CVP, and 10 + 2.5 shows hy-LHBs Ag 10 µg and HBc Ag 2.5 µg plus CVP #00 and without CVP (CVP-). “None” indicates sera from non-treated mice. PS is anti-HBc mouse sera or anti-hy-LHBs mouse sera, and b is the background only containing PBS-. Vertical bars indicate S.D.

**Figure 3 vaccines-13-00464-f003:**
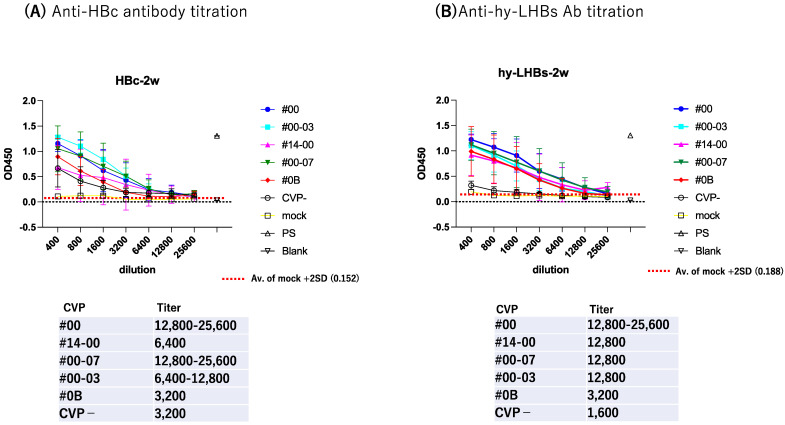
Serial dilution of mouse Ab against HBc (**A**) and hy-LHBs (**B**) and detection by ELISA. Mice were immunized with HBV antigen and various CVP excipients and were analyzed after 2 weeks. “Mock” represents non-treatment control, and “PS” indicates anti-HBc or hy-LHBs mouse sera. Blank contains only PBS-. Vertical bars indicate S.D.

**Figure 4 vaccines-13-00464-f004:**
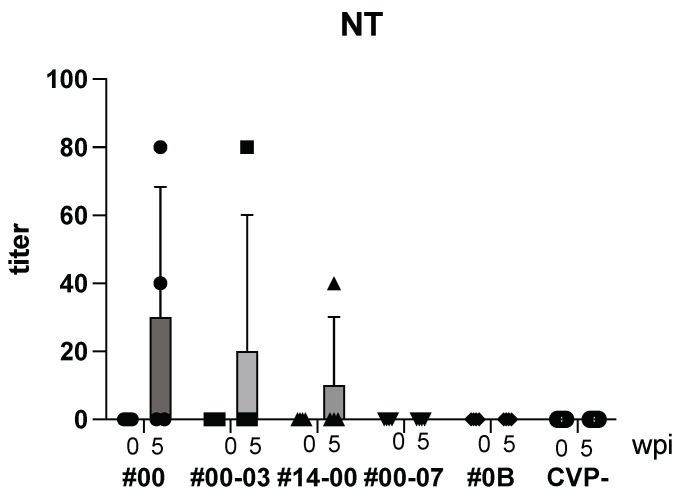
Detection of NT Ab in BALB/c mice with hy-LHBs, HBc, and antigen with CVPs. CVP- is hy-LHBs and HBc Ag 10 μg each without CVP. wpi: weeks post immunization. Vertical bars are S.D.

**Figure 5 vaccines-13-00464-f005:**
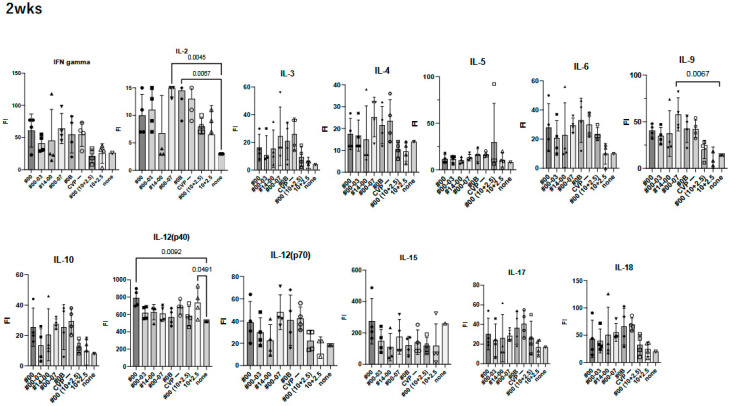
Comprehensive analysis of cytokines and chemokines by BioPlex using 2 weeks post-immunized mouse sera. Mice were immunized with hy-LHBs + HBc (10 μg each) in the presence of CVP #00, #00-03, #14-00, #00-07, #0B, or no CVP (CVP-), hy-LHBs and HBc Ag (10 and 2.5 μg, respectively) in the presence of #00 or no CVP, or had no treatment (control, none). FI is fluorescent intensity and vertical bars indicate S.D.

**Figure 6 vaccines-13-00464-f006:**
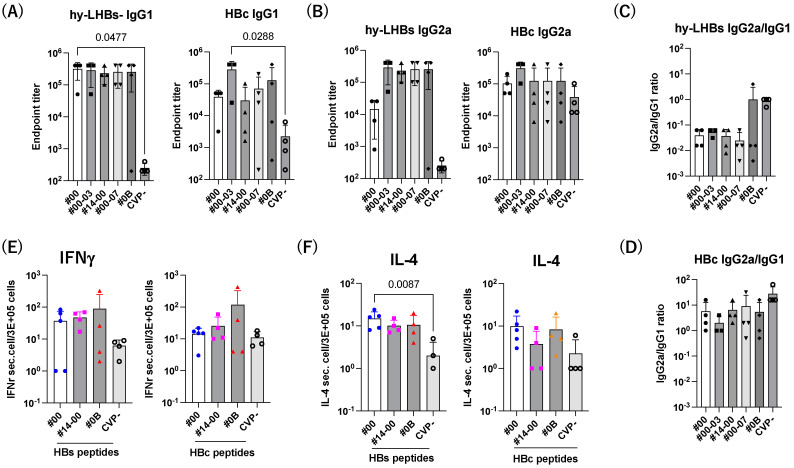
Th1/Th2 immune response to hy-LHBs and HBc. (**A**) IgG1 and (**B**) IgG2 responses against hy-LHBs (left) and HBc (right), which were measured by endpoint titers. (**C**,**D**) Th1/Th2 response was calculated by the ratio of IgG2a/IgG1 using the endpoint titers. (**E**) The ELISpot assay of IFN-γ and IL-4 (**F**) in splenocytes from mice receiving HBV antigens and various CVP excipients following hy-LHBs and HBc peptide treatment as indicated under lines. Mouse splenocytes were isolated and cultured for 24 h under a range of conditions. Mice were immunized with hy-LHBs and HBc antigens with CVP Type #00, #14-00 and #0B formulations. Vertical bars indicate S.D.

**Figure 7 vaccines-13-00464-f007:**
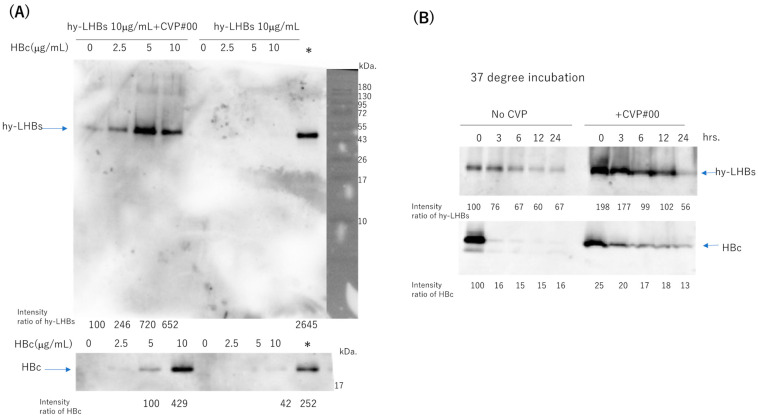
Stability of hy-LHBs antigen and HBc antigen in the presence or absence of a CVP excipient (CVP Type #00 formulation). (**A**). hy-LHBs antigen (10 μg/mL) was mixed with 0, 2.5, 5, or 10 μg/mL of HBc Ag with or without CVP#00 and incubated at 4 °C for 1 week to visualize the hy-LHBs antigen (upper) and HBc antigen (lower) in WB analysis. The molecular weight marker is also shown in the right lane. *: hy-LHBs + HBc, (10 μg/mL each) keep at −30 °C. (**B**) Stability of hy-LHBs antigen and HBc antigen (10 μg/mL) with or without CVP#00 was examined after incubation at 37 °C for 0, 3, 6, 12, and 24 h.

## Data Availability

Data will be made available upon reasonable request.

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
