# Peer review of "Immunogenicity of an Intranasal Dual (Core and Surface)-Antigen Vaccine Against Hepatitis B Virus Enhanced by Carboxyl-Vinyl Polymer Excipients"

_vaccines, 2025, doi:10.3390/vaccines13050464_

Round 1

Reviewer 1 Report

Comments and Suggestions for Authors

Despite intensive research in the field, attempts to develop effective therapeutic hepatitis B vaccines have not been successful so far. Novel strategies are needed to produce an efficient therapeutic HBV vaccine that is able to elicit a potent anti-viral immunity and achieve efficient virological suppression. This study describes the use of carboxyl-vinyl polymers (CVP) as mucoadhesive excipients to enhance stability and immunogenicity of HBV antigens for intranasal administration of HBV vaccines with therapeutic potential. There are several issues that need to be addressed to improve the quality of manuscript, specifically:

Major:

-Figure 2, legend and the corresponding text section in Results, please indicate which antibodies (IgG or IgA) were measured? If the IgG levels are shown, considering the administration route of these vaccine candidates and the aim to stimulate the mucosal immunity, IgA levels need to be determined and included in the analysis.

Figure 3: Improved antibody titers following vaccination in the presence of CVP, does not necessarily mean these antibodies are of the desired quality, i.e. virus-neutralizing. The authors should at least attempt to check the binding of these antibodies to HBV particles, not only to the antigens they used in vaccination, ideally perform the HBV neutralization assays in vitro.

Figure 2: How many mice were included per experimental group? If data is shown per individual animal, why is the number of points shown in the datasets of each panel so variable (from 1 to 4)?

Figure 3: Why did the authors switch to a different antigen to determine the IgA-based immune response (as opposed to the combinations used in Fig. 2)? There is no explanation provided in the manuscript; similarly for the choice of particular CVP formulations for this antigen, what was this selection based on?

Figures 5 and 6: IFN-γ and IL-4 (as well as IL-5, IL-6, IL-10) are the signature cytokines for Th1 (cellular immune response) and Th2 cell activation (humoral immune response), respectively. Addition of CVP does not appear to significantly change these responses in vaccinated mice. What is the biological relevance of these results? For a better insight into the nature of the immune response triggered by these antigen formulations, the authors could quantify the levels of IgG isotypes, IgG1 and IgG2a as a measure of humoral and cellular immune response activation, respectively.

Figure 7: The blot shows the amounts of antigens per sample, but there is no indication as to the total protein concentration, therefore the results suggest that the antigen stability in the presence of CVP is actually correlated with the protein concentration, rather than the antigen ratio (Discussion, lines 353-360).

Discussion: some claims regarding the efficacy of CVP formulations are not supported by the presented data, for example, in lines 318-324. As shown in Fig 2, there is no significant increase of the anti-HBc IgG response in the presence of either CVP formulation used over the course of immunization.

Minor:

-Confusing statements; lines 93-94, the authors state that: “Immunogenicity has been characterized for HBc and HBs-Lh antigens when delivered intranasally with a CVP excipient, and then further, lines 97-98, they add: ”However, immunogenicity has not been evaluated for a vaccine with HBs-Lh antigen mixed with a CVP excipient”. Do the authors mean that the HBc and HBs-Lh combination has not been evaluated?

-Page 3, Experimental vaccines formulation section: please clarify the entry #5 in the table. What does this formulation contain?

-Lines 103-104: the authors state that a combination of HBc and HBs-S/HBs-L antigen, in the presence or absence of a CVP excipient administered with intranasal administration; however, no HBs-S antigen is listed among the formulations shown in the table (page 3). Instead the HBsAg alone (i.e. not in combination with HBc) is used in a subsequent immunisation experiment (Fig. 4).

-The experimental groups shown in Fig 1 do not entirely correspond to the vaccine formulations shown in the table (page 3). Are the amounts of antigens  in the table indicated as per dose?  Antigen doses are confusing in figure legends, are these micrograms (Fig. 2) or milligrams (Figs 4 and 5?)

-In mammalian cells, the HBV-L protein has one N-linked glycan in the S domain.  What is the origin of the HBs-Lh antigen used in this study and why is there no visible glycosylated form (Fig. 7)?

-Figure legends are not very informative: the numbers shown in graphs are likely p values. If this assumption is correct, it would be more meaningful to indicate the levels (intervals) of significance rather than various numbers for each intergroup comparison and mark as “ns” statistically non-relevant differences. What are the vertical bars on each graph? Please clarify.

Author Response

Despite intensive research in the field, attempts to develop effective therapeutic hepatitis B vaccines have not been successful so far. Novel strategies are needed to produce an efficient therapeutic HBV vaccine that is able to elicit a potent anti-viral immunity and achieve efficient virological suppression. This study describes the use of carboxyl-vinyl polymers (CVP) as mucoadhesive excipients to enhance stability and immunogenicity of HBV antigens for intranasal administration of HBV vaccines with therapeutic potential. There are several issues that need to be addressed to improve the quality of manuscript, specifically:

Major:

-Figure 2, legend and the corresponding text section in Results, please indicate which antibodies (IgG or IgA) were measured? If the IgG levels are shown, considering the administration route of these vaccine candidates and the aim to stimulate the mucosal immunity, IgA levels need to be determined and included in the analysis.

Response: Thank you for your significant comments. We have indicated as IgG (line **). We have measured IgA based on the reviewer’s instruction Fig.2(E).

Figure 3: Improved antibody titers following vaccination in the presence of CVP, does not necessarily mean these antibodies are of the desired quality, i.e. virus-neutralizing. The authors should at least attempt to check the binding of these antibodies to HBV particles, not only to the antigens they used in vaccination, ideally perform the HBV neutralization assays in vitro.

Response: We have performed virus-NT test (Fig.4).

Figure 2: How many mice were included per experimental group? If data is shown per individual animal, why is the number of points shown in the datasets of each panel so variable (from 1 to 4)?

Response: We have used 4 mice per each experiment group. During the course of immunization, earlier periods (2wks) show higher variation, which might be due to the individual differences.

Figure 3: Why did the authors switch to a different antigen to determine the IgA-based immune response (as opposed to the combinations used in Fig. 2)? There is no explanation provided in the manuscript; similarly for the choice of particular CVP formulations for this antigen, what was this selection based on?

Response: We have changed IgA response results on the same Ag, HBs-Lh and HBc (Fig.2E).

Figures 5 and 6: IFN-γ and IL-4 (as well as IL-5, IL-6, IL-10) are the signature cytokines for Th1 (cellular immune response) and Th2 cell activation (humoral immune response), respectively. Addition of CVP does not appear to significantly change these responses in vaccinated mice. What is the biological relevance of these results? For a better insight into the nature of the immune response triggered by these antigen formulations, the authors could quantify the levels of IgG isotypes, IgG1 and IgG2a as a measure of humoral and cellular immune response activation, respectively.

Response: Thank you for your insightful comments. We measured end point titers of IgG1 and IgG2a (Fig.6A-D).and were higher with CVPs than without. Higher IgG1 response was induced to hy-LHBs and higher IgG2a was induced to HBc by CVPs.

Figure 7: The blot shows the amounts of antigens per sample, but there is no indication as to the total protein concentration, therefore the results suggest that the antigen stability in the presence of CVP is actually correlated with the protein concentration, rather than the antigen ratio (Discussion, lines 353-360).

Response: Thank you for your significant instruction. We have added the detailed stability test methods (line **) and modified Fig.7 to indicate by protein concentration.

Discussion: some claims regarding the efficacy of CVP formulations are not supported by the presented data, for example, in lines 318-324. As shown in Fig 2, there is no significant increase of the anti-HBc IgG response in the presence of either CVP formulation used over the course of immunization.

Response: After 2 weeks of immunization, CVP #00-03 induced significant upregulation of anti-HBc Ab comparing with CVP- (Fig.2A). Also amount of IgG1 was significantly increased by CVP#00-03 (Fig.6C).

Minor:

-Confusing statements; lines 93-94, the authors state that: “Immunogenicity has been characterized for HBc and HBs-Lh antigens when delivered intranasally with a CVP excipient, and then further, lines 97-98, they add: ”However, immunogenicity has not been evaluated for a vaccine with HBs-Lh antigen mixed with a CVP excipient”. Do the authors mean that the HBc and HBs-Lh combination has not been evaluated?

Response: We apologize this mistake. We have removed this sentence in lines 97-98.

-Page 3, Experimental vaccines formulation section: please clarify the entry #5 in the table. What does this formulation contain?

Response: The entry #5, CVP#0B is approximately 5-fold higher viscosity than #00.

-Lines 103-104: the authors state that a combination of HBc and HBs-S/HBs-L antigen, in the presence or absence of a CVP excipient administered with intranasal administration; however, no HBs-S antigen is listed among the formulations shown in the table (page 3). Instead the HBsAg alone (i.e. not in combination with HBc) is used in a subsequent immunisation experiment (Fig. 4).

Response: We have replaced Fig.4 with Fig.2E and HBs-S antigen to HBs-Lh antigen. As a result, we are not necessary to modify the table in page 3.

-The experimental groups shown in Fig 1 do not entirely correspond to the vaccine formulations shown in the table (page 3). Are the amounts of antigens in the table indicated as per dose?  Antigen doses are confusing in figure legends, are these micrograms (Fig. 2) or milligrams (Figs 4 and 5?)

Response: We apologize these errors. We have modified the information in table to be consistent with Fig.1. Milligrams in Fig.4 and 5 were modified to micrograms.

-In mammalian cells, the HBV-L protein has one N-linked glycan in the S domain.  What is the origin of the HBs-Lh antigen used in this study and why is there no visible glycosylated form (Fig. 7)?

Response: The HBs-L protein used in this study was expressed in yeast and purified by Beacle Co. The HBs-Lh protein is composed of PreS1, PreS2 and S proteins and expected molecular weight is approximately 42 kDa. Its glycosylation in one site gave increased molecular weight, 45 kDa.

-Figure legends are not very informative: the numbers shown in graphs are likely p values. If this assumption is correct, it would be more meaningful to indicate the levels (intervals) of significance rather than various numbers for each intergroup comparison and mark as “ns” statistically non-relevant differences. What are the vertical bars on each graph? Please clarify.

Response: In line with the reviewer's comments, we modified to indicate that p-values less than 0.05 (significant) were indicated. Also, the vertical bars were S.D. and indicated in figure legends.

Reviewer 2 Report

Comments and Suggestions for Authors

The authors compared immunogenicity of various CVP excipients with a dual-antigen HBV vaccine in mice via intranasal immunization by measuring both humoral and cellular immune responses. The authors also evaluated the stability of experimental vaccines and found HBc is required for stable formulations with HBs-Lh. Overall, experiments are well performed and analyzed but several descriptions are not clear. Specific comments follow.

Major points:

  1. ABSTRACT: HBs is not a dual-antigen vaccine (line24), “2.5 μg HBs-Lh” should be “HBc” (line 29), “A significant increase in IgA antibody 29 production was observed with HBs Ag and CVP#00, #0B, and #14-00.” is wrong (line 29) and “#00” is missing for “MIP1α” (line 31).
  2. Please justify to analyze IgA antibody as HBV vaccine efficacy. Is there any reference describing the role of IgA antibody in protection?
  3. Line 116: Please add the details of the experiment shown in Figure 4.
  4. Figure 1: Please change title like “Schematic study schedule” and correct the position of black allow (Should be week 5) and add red allow (Booster) under week 4. Also please indicate weekly blood collection and use same group numbers to section 2.3 for easier reading.
  5. Line 132: Please delete “livers, kidneys, hearts, lungs, and intestines” as I don’t see the results.
  6. Line 143: “IgG” should be “antibody” as you also measured IgA. The manuscript is confusing because you mixed two experiments used different antigens in one. Can the authors measure IgA antibodies for Figure 2 samples and delete Figure 4?
  7. Lines 160-161: Please explain why the authors used different wave-length to read.
  8. Figure 2 (B): In HBc-3w graph, please delete two comparisons as both of them are not significant, for consistency.
  9. Figure 2 (C): In HBc-4w graph, which groups were compared?
  10. Figure 4 (A): Please replace “Measure IgA” by “Bleeding” and make it clear which antigen was used. Was it HBs-Lh? What do you mean by “HBs antigens” but not “HBs antigen”?
  11. Figure 6: Please remove ConA from the graphs to see the statistical differences against CVP- group.
  12. Line 298: Please show ELISA results for week 5 samples to see the boosted immune response.
  13. Lines 298-300 and 334-337: I don’t see any significant increase of cells producing IFN-γ and IL-4 in immunized mice with CVP formulations (Figure 6).
  14. Line 353: Can the authors show SDS-PAGE of HBs-Ln antigen stored at 4°C for several days to show the degradation? Lost of HBs-Ln bands could be due to adsorption of antigen to the tube rather than degradation of antigen. Furthermore, to test the stability of antigens at 25 to 37°C to mimic the temperature of nostril will be interesting.

Minor points:

  1. Line 70: Please spell out “CHB”.
  2. Lines 153 & 184: What do you mean by “0.5% PBS”?
  3. Line 153: Please insert “(PBST)” after “Tween-20.
  4. Line 184: Please replace “0.5% PBS with 0.05% Tween-20” by “PBST”.
  5. Figure 4: “10mg” should be “10 µg”.

Author Response

Reviewer 2:

The authors compared immunogenicity of various CVP excipients with a dual-antigen HBV vaccine in mice via intranasal immunization by measuring both humoral and cellular immune responses. The authors also evaluated the stability of experimental vaccines and found HBc is required for stable formulations with HBs-Lh. Overall, experiments are well performed and analyzed but several descriptions are not clear. Specific comments follow.

Major points:

  1. ABSTRACT: HBs is not a dual-antigen vaccine (line24), “2.5 μg HBs-Lh” should be “HBc” (line 29), “A significant increase in IgA antibody 29 production was observed with HBs Ag and CVP#00, #0B, and #14-00.” is wrong (line 29) and “#00” is missing for “MIP1α” (line 31).

Response: Thank you for your instruction. We have updated in line with the reviewer's comments.

  1. Please justify to analyze IgA antibody as HBV vaccine efficacy. Is there any reference describing the role of IgA antibody in protection?

Response: There are a few reports showing the effect of IgA in HBV vaccine (Vaccines 9(2) 101, 2021; Vaccine 25, 577-584, 2007 etc.).

  1. Line 116: Please add the details of the experiment shown in Figure 4.

Response: In line with the reviewer’s comments, we have removed Figure 4.

  1. Figure 1: Please change title like “Schematic study schedule” and correct the position of black allow (Should be week 5) and add red allow (Booster) under week 4. Also please indicate weekly blood collection and use same group numbers to section 2.3 for easier reading.

Response: We have modified Figure 1, based on the reviewer’s instruction.

  1. Line 132: Please delete “livers, kidneys, hearts, lungs, and intestines” as I don’t see the results.

Response: We have deleted based on the reviewer’s instruction.

  1. Line 143: “IgG” should be “antibody” as you also measured IgA. The manuscript is confusing because you mixed two experiments used different antigens in one. Can the authors measure IgA antibodies for Figure 2 samples and delete Figure 4?

Response: We have replaced IgG to antibody and Figure 4 with Figure 2E, based on the reviewer’s instruction.

  1. Lines 160-161: Please explain why the authors used different wave-length to read.

Response: It is because of the difference of detection substrate, but we removed as we deleted Figure 4.

  1. Figure 2 (B): In HBc-3w graph, please delete two comparisons as both of them are not significant, for consistency.

Response: We have deleted, based on the reviewer’s instruction.

  1. Figure 2 (C): In HBc-4w graph, which groups were compared?

Response: It was a result of One-way Anova, and we have removed as it is confusing.

  1. Figure 4 (A): Please replace “Measure IgA” by “Bleeding” and make it clear which antigen was used. Was it HBs-Lh? What do you mean by “HBs antigens” but not “HBs antigen”?

Response: We used HBs antigen for old Figure 4 and it was removed, in line with reviewer’s request.

  1. Figure 6: Please remove ConA from the graphs to see the statistical differences against CVP- group.

Response: We have removed ConA in line with the reviewer’s comments.

  1. Line 298: Please show ELISA results for week 5 samples to see the boosted immune response.

Response: We showed ELISA results for week 5 in Figure 2D.

  1. Lines 298-300 and 334-337: I don’t see any significant increase of cells producing IFN-γ and IL-4 in immunized mice with CVP formulations (Figure 6).

Response: Only the significant increase by CVP#00 is observed in IL-4 secretion (p=0.0087) in Figure 6B.

  1. Line 353: Can the authors show SDS-PAGE of HBs-Ln antigen stored at 4°C for several days to show the degradation? Lost of HBs-Ln bands could be due to adsorption of antigen to the tube rather than degradation of antigen. Furthermore, to test the stability of antigens at 25 to 37°C to mimic the temperature of nostril will be interesting.

              Response: We are using low protein bound tube for experiment, therefore, we are not thinking that these decrease is due to the adsorption to the tube. Based on your instruction, we have examined the stability of HBs-Lh and HBc Ag at 37 degree in Fig.7B.

Minor points:

  1. Line 70: Please spell out “CHB”.

Response: We have spell out as chronic hepatitis B for CHB.

  1. Lines 153 & 184: What do you mean by “0.5% PBS”?

Response: We are sorry about this mistake. We have removed 0.5%.

  1. Line 153: Please insert “(PBST)” after “Tween-20.

Response: We have inserted PBST.

  1. Line 184: Please replace “0.5% PBS with 0.05% Tween-20” by “PBST”.

Response: We have removed PBST.

  1. Figure 4: “10mg” should be “10 µg”.

Response: We have removed Figure 4.

Round 2

Reviewer 1 Report

Comments and Suggestions for Authors

The authors have made a significant effort to address the reviewer's comments and concerns and the current version of the manuscript has considerably improved. Some assays are still not optimally performed (e.g, the HBV "neutralization" experiment did not investigate the mice pre- immune sera to rule out potential non-specific interactions). However, I will endorse the publication of the manuscript in the current form, based on the complementary data provided.

Author Response

The authors have made a significant effort to address the reviewer's comments and concerns and the current version of the manuscript has considerably improved. Some assays are still not optimally performed (e.g, the HBV "neutralization" experiment did not investigate the mice pre- immune sera to rule out potential non-specific interactions). However, I will endorse the publication of the manuscript in the current form, based on the complementary data provided.   Response: Thank you for your insightful comments. As suggested, we have included the pre-immune serum (day 0) in the revised manuscript. The results showed no reactivity, indicating the absence of non-specific interactions. We sincerely appreciate your comment, which has helped us to further improve the manuscript.

Reviewer 2 Report

Comments and Suggestions for Authors

The authors have substantially revised their manuscript and addressed all of the queries.

Minor issues:

  1. Figures 1&2: “µ” are not shown properly in PDF file.
  2. Line 215: 37°C is missing.
  3. Figure 6: Please use same Y-axis scale for easier comparison and Figure 6F; Y-axis should be Log10 scale for consistency.

Author Response

The authors have substantially revised their manuscript and addressed all of the queries.

Response: Thank you for your insightful comments.

Minor issues:

  1. Figures 1&2: “µ” are not shown properly in PDF file.

Response: We are sorry about this inconvenience and have updated.

  1. Line 215: 37°C is missing.

Response: We apologize for this oversight and have updated it in the revised manuscript.

  1. Figure 6: Please use same Y-axis scale for easier comparison and Figure 6F; Y-axis should be Log10 scale for consistency.

Response: We sincerely regret the inconvenience and have accordingly revised Figure 6F.